# Monolithic Integration of GaN-Based Transistors and Micro-LED

**DOI:** 10.3390/nano14060511

**Published:** 2024-03-12

**Authors:** Honghui He, Jinpeng Huang, Tao Tao, Ting Zhi, Kaixin Zhang, Zhe Zhuang, Yu Yan, Bin Liu

**Affiliations:** 1Jiangsu Provincial Key Laboratory of Advanced Photonic and Electronic Materials, School of Electronic Science and Engineering, Nanjing University, Nanjing 210093, China; 2College of Electronic and Optical Engineering & College of Flexible Electronics, Nanjing University of Posts and Telecommunications, Nanjing 210023, China; 3National and Local United Engineering Laboratory of Flat Panel Display Technology, College of Physics and Information Engineering, Fuzhou University, Fuzhou 350100, China; 4Fujian Science & Technology Innovation Laboratory for Optoelectronic Information of China, Fuzhou 350100, China

**Keywords:** micro-LED, display, monolithic integration

## Abstract

Micro-LED is considered an emerging display technology with significant potential for high resolution, brightness, and energy efficiency in display applications. However, its decreasing pixel size and complex manufacturing process create challenges for its integration with driving units. Recently, researchers have proposed various methods to achieve highly integrated micro-structures with driving unit. Researchers take advantage of the high performance of the transistors to achieve low power consumption, high current gain, and fast response frequency. This paper gives a review of recent studies on the new integration methods of micro-LEDs with different types of transistors, including the integration with BJT, HEMT, TFT, and MOSFET.

## 1. Introduction

### 1.1. Emerging of Micro-LED Display Technology

Half a century ago, plasma display (PDP) technology was a popular display technology for time [1,2]. PDP creates an image by exciting a phosphor to emit light through the discharge of ionized gases. Due to its high energy consumption and high costs, PDP was gradually replaced by other display technologies. Subsequently, liquid crystal display (LCD) monitors, which can regulate the brightness and color of pixels by controlling the arrangement of liquid crystal molecules and the polarization of light to display the image, became popular as the dominant display technology for nearly half a century [3]. Now, organic light-emitting diode (OLED) display technology is an important display technology, which uses the electroluminescence of organic materials to produce images [4]. It does not require a backlight and therefore has a higher contrast ratio, a wider viewing angle, and a faster response time. OLED consists of a stack of organics (ITO) sandwiched between a cathode and anode, where electrons and holes are injected from the electrodes into the organic layer for compounding and light emission. In recent years, the development of OLEDs has centered around two main areas: increasing the optical output power and thinning the device architecture. Currently, researchers are mainly using ordered structures such as (surface mounted) metal–organic framework (SUR) MOFs to regulate electron–hole mobility. This approach can enhance the optical output power while simplifying the structure of OLED devices. They also have great potentials for applications in the field of wearable devices and flexible displays.

In recent years, newly developed micro-LED displays have attracted a lot of attention. A micro-LED is a specially designed LED with an extremely small pixel size, which is capable of achieving high-definition, high-brightness, and high-contrast display performance [5,6,7,8,9,10]. The high efficiency of GaN-based-LEDs could offer low power consumption compared to other techniques. Currently, micro-LED displays have demonstrated great potential in near-to-eye applications, but they are still facing many challenges in terms of manufacturing and integration. Wearable devices, for example, virtual reality (VR) and augmented reality (AR) devices, high-definition large displays, and other areas, require extremely high performance from both micro-LEDs and driving units as well. With the continuous development and maturity of the micro-LED technique, it is expected that micro-LED displays will find a place in the future market and become the next dominant display technology.

Micro-LED technology was firstly proposed by Prof. Jiang and Prof. Lin in Texas Tech University. Generally, it is widely accepted that micro-LEDs do not exceed 50 microns in size. For different display applications, there are two main trends. One of the directions is that which Sony is focusing on, which is small-pitch, high-resolution, large-size indoor/outdoor displays. The other is the wearable devices (such as AR/VR and Watch) launched by Apple, which require high resolution, portability, low power consumption, and high brightness. It is important to make the use of the many advantages of micro-LEDs: fast response time, high brightness, wide viewing angle, high dynamic range, low power consumption, and long lifetime. The history of display technology is shown in Figure 1.

### 1.2. Nanostructure for GaN-Based Micro-LED

Micro-LEDs are gradually replacing LCDs and OLEDs as the requirements for display resolution, brightness, and stability increase, but the EQE of quantum-well LEDs decreases as the size of the device is drastically reduced to around 100 µm. Quantum well LEDs will introduce a large number of dislocations and defects due to the ICP etching table, which will induce non-radiative surface bonding. There will be a certain lattice mismatch between InGaN and GaN in the quantum well, which means that InGaN quantum well LEDs face certain difficulties in efficiently emitting red and green light. To overcome this difficulty, nanowires/nanorods can be deposited on the substrate. Through selective growth, the size and morphology of the nanowires/nanorods can be precisely controlled to achieve the effect of changing the emission wavelength [11,12,13,14]. By using SAG to produce highly uniform nanowires of the same polarity, the inhomogeneity associated with spontaneous nanowires can be eliminated [15]. Such dislocation-free nanostructures can further enhance the performance and functionality of micro-LEDs, including narrow spectral linewidths and highly directional and stable emission.

### 1.3. Major Challenges in Micro-LED Integration Process

There are two challenges in the micro-LED integration process that have received a lot of attention from researchers, which are RGB integration and driving unit integration. Firstly, it is important to achieve RGB full-color micro-LED display. Therefore, many methods have been proposed. In this paper, three newly developed methods are introduced.

Three-color stacking: The three-color stacking technology achieves the full color display by controlling each vertically stacked red, green, and blue micro-LED, as shown in Figure 2a. The stacking structure allows individual pixels to occupy less space, thereby achieving a higher pixel density per unit area, meeting the requirements for application in small-size micro-display devices and also having wide applications in the display field [16,17,18]. Therefore, it is considered to be one of the key technologies for achieving high resolution and high brightness in the future, if it can overcome the difficulty in chip manufacturing and packaging technology. After all, it needs a complex optical design, precise spacing, and accurate alignment of different layers. Color uniformity and consistency, as well as interconnections with driving units, are all challenges that need to be solved for three-color stacking.

Quantum dots: Integrating Micro-LEDs with quantum dots could be an easy way to achieve a broader color gamut and higher contrast display by the way of color conversion. Most quantum dots are spherical or quasi-spherical semiconductor nanostructures that can be excited to emit light of specific wavelengths [19,20]. In most case, the micro-LED chip emits blue light and the quantum dots absorb the blue light and re-emit red and green light, resulting in full-color display. By controlling the size and composition of the quantum dots, the color of the emitted light can be tuned, achieving a wider color gamut [21]. Although wider color gamut, better color display effects, higher brightness, and higher display quality are all advantages of quantum dots, the stability of quantum dots and the difficulty of preparing quantum dot thin films are urgent problems to be solved. Nevertheless, the unique optoelectronic properties of quantum dots provide one more solution for full-color micro-LED displays.

Mass transfer: The primary goal of mass transfer technology is to transfer a large number of micro-LED pixels from the growth substrate to the target display substrate with high precision and efficiency in a short time [22]. It has been adopted in large-size display applications. In fact, mass transfer technology plays a crucial role in micro-LED fabrication. It enables abundant, high-precision micro-LED chip transfer, providing key technical support for achieving high-resolution and high-brightness displays. However, it is important for mass transfer technology to reduce manufacturing costs and improve production efficiency.

The mass transfer process can be divided into two main steps: separation and transfer [23]. In the separation stage, micro-LED pixels need to be detached from the wafer by methods such as mechanical peeling, chemical peeling, or laser peeling. In the transfer stage, the separated micro-LED chips are transferred to the target substrate through adsorption, molecular electrostatic attraction, or current-driven methods as shown in Figure 2c. During the mass transfer process, multiple factors need to be considered, such as the adhesion force between the chip and the substrate, temperature control, and alignment accuracy [24]. Moreover, mass transfer technology needs to address the uniformity and consistency issues of micro-LED chips, ensuring that the transferred chips have stable performance and optoelectronic properties.

There are several ways to drive micro-LEDs, the most commonly used being TFT and CMOS driver circuits as shown in Figure 3. The discrete driver allows each micro-LED to be individually controlled by driver circuits to achieve color, brightness, etc.

TFT driver circuit: The TFT (Thin-Film Transistor) driver circuit is an active matrix driver circuit commonly used in current displays. It is widely used in various types of flat panel displays, such as LCD TVs, laptop computer displays, mobile phone screens, and so on. The TFT driver circuit usually consists of a driver chip, a TFT array, and power/signal lines. The driver chip is the core part, receiving external signals and controlling the conduction and turn-off of the TFT [25]. The TFT array works as a current switch to control the brightness and color of each pixel by changing the conduction state. The transistors are usually made of monocrystalline or polycrystalline silicon and have high current driving capability and low switching voltage. Power and signal circuits are used to provide power and transmit control signals. Multiple supply voltages and clock signals are usually required to ensure that the thin-film transistors operate properly [26].

In recent years, amorphous oxide semiconductor materials, represented by IGZO, have received extensive attention for their rich properties. Compared to conventional a-Si TFTs, it has the advantages of good uniformity, simple manufacturing, low manufacturing cost, and high carrier mobility [27,28,29]. Besides, IGZO is a wider energy bandgap ~3 eV material with good transparency and has become one of the alternative materials for transparent displays.

CMOS driver circuits: The CMOS driver circuit is an active matrix driver circuit commonly used in LCDs and other electronic devices [30,31]. CMOS driver circuits typically include a driver chip, p-MOS and n-MOS transistor arrays, and power and signal lines. Therefore, they have low power consumption and high reliability. The driver chip receives external signals and controls the conduction and turn-off of the transistors. CMOS regulates the current and voltage at the pixel points by controlling the on and off states of the transistors to achieve brightness and color changes in the display. Power and signal lines are used to supply power and transmit control signals.

Compared to TFT drivers, the major advantages of CMOS driver circuits are high integration, low power consumption, and high reliability [32]. Thus, they are very suitable for near-to-eye display applications such as AR/VR, smart watches, smartphones, head-up displays (HUDs) and others. Micro-LEDs are changing fast; therefore, the design and manufacturing process of CMOS driver circuits are being improved to meet the higher demand of quality and integration with low cost.

Difficulties for discrete driver circuits: With today’s demands for high resolution display with much smaller micro-LEDs, the traditional TFT driver circuit presents a number of challenges. Typically, the pixel size of a TFT driver is 100 μm. It is hard to further reduce the size of TFTs, which will result in low drive efficiency and high power consumption for current regulation. CMOS drivers could achieve nano-sized control units, supporting the high resolution demand, but the high cost, difficulty in the bonding process, and RGB integration need to be settled. In order to overcome these problems, many research teams have explored new integration methods of driver units and micro-LEDs to improve the electrical and optical characteristics. In this paper, we focus on the new on-chip integration of GaN-based driver units with micro-LEDs [26]. Conventional TFT driver circuits have a threshold voltage of around 2.9 V and a mobility of 10.5 cm^2^/(V·S) [25]. However, MOSFETs integrated with micro-LEDs can reach a threshold voltage of 0.8 V [33], which means that the lower threshold voltage in low-power devices and systems allows the transistors to operate at lower voltages for switching, which reduces the power consumption of the entire system. When we use a HEMT integrated with a micro-LED, the carrier mobility can reach 1810 cm^2^/(V·S) [34], which means that higher carrier mobility leads to higher current drive capability for faster switching speeds, which is very important for high-speed logic circuits and high-frequency RF devices.

## 2. Innovation and Development of On-Chip Integration Technology

Integration technology plays a vital role in micro-LED display. Micro-LEDs offer higher injection current density, smaller pixel size, higher brightness, and longer lifetimes than conventional display. Therefore, integration technology is very critical for the development of high-performance micro-LED displays [35,36]. Here, it is important to obtain high performance in current, voltage, and signal power in GaN-based transistors. Integrated devices are more difficult to fabricate compared to individual devices, but the cost of manufacturing is not significantly higher when all modules are made on the single chip. Currently, on-chip integration still faces many challenges in micro-LED displays. For example, during the manufacturing process, a high-temperature treatment of ~800 °C is required to form Ohmic contacts, which would exceed the thermal budget of the LED device, resulting in a degradation of device performance. There are some difficulties and challenges in the process of integrating different transistors with micro-LEDs. Because micro-LEDs are very small, any tiny defect can cause the entire display to fail, so how to improve the yield rate is a challenge that needs to be solved. The transistors and micro-LEDs must be precisely matched in size to ensure that each micro-LED is properly controlled, which requires very precise equipment and processes. Although micro-LEDs are inherently more energy efficient, they can generate a lot of heat when operating at high brightness. Effective thermal management solutions are therefore required to prevent overheating from affecting the performance of the device. In order to enhance the reliability of integrated devices, researchers generally take some methods to improve the reliability of micro-LEDs, such as reducing the defect density of epitaxial growth, designing efficient device structures, and developing effective thermal management [9].

In this paper, we focus on the integration of different kinds of GaN-based drivers and micro-LEDs. We report on the integration of GaN-based Bipolar Junction Transistors (BJTs), MOS Field Effect Transistors (MOSFETs), and High Electron Mobility Transistors (HEMTs) with micro-LEDs in the same chip, reducing the size, improving efficiency, reducing power consumption, and also improving system stability.

### 2.1. BJT-Integrated Micro-LED

An ideal bipolar transistor (BJT) consists of two pn structures with different doping concentrations. In 2022, S. Hao et al. from Fuzhou University simulated a new type of light-emitting triode, as shown in Figure 4a [37]. One GaN-based LED was vertically integrated with one npn-type BJT on the same chip, where the cathode of the LED and the collector of the BJT share the same n-GaN layer. This vertically integrated structure provides the ability of acquiring higher-resolution display compared to HEMT and MOSFET integration. The optical output power of the integrated device can be regulated by a smaller base-level current. The cut-off frequency can be raised up to 80 MHz and more. The luminous intensity is closely related to the frequency, amplitude, and duty cycle of the pulse signal, proving a stronger capability of controlling the device. However, there are still many difficulties in the process to be solved. The difference in epitaxial temperature and lattice mismatch between P-GaN and N-GaN will result in a poor device performance.

From Figure 4c, it can be seen that an increase in the doping level of the emitter region will improve the current gain. As the doping level of the emitter region increases, more electrons can be injected into the base region, and more electrons will be collected in the collector region. From Figure 4b, it can also be found that the doping level improved the current gain, thus enhancing the overall luminescence intensity of the device. The doping level in base region should not be too low. When the doping level is 5 × 10^17^ cm^−3^, the output characteristic is close to the ideal state. The increase in the doping level in transition region will increase the current gain and also reduce the turn-on voltage of the integrated LED. Theoretical models play a crucial role in supporting and interpreting experimental results. They provide a framework within which experimental observations can be understood and explained. This simulation provided later researchers with theoretical experience on the integration of BJTs with MOSFETs, including the doping concentration of each GaN layer and the frequency response characteristics of the whole device.

In 2021, Fu et al. at Hong Kong University proposed a PNP BJT and LED integrated device as shown in Figure 5a. The devices are fabricated on a typical blue-emitting InGaN/GaN LED wafer grown on a c-plane patterned sapphire substrate (PSS) by metalorganic vapor-phase epitaxy (MOVPE) [38]. Fabrication of the device begins with electrically isolating the mesa by a Cl_2_-based inductively-coupled plasma (ICP) etch down to the sapphire substrate. The base region is then ICP etched down to the n-GaN layer. Two types of LEBJTs with different emitting areas were designed and reported: one with a larger emitting area for optoelectronic signal conversion, and the other one with a smaller emitting area for improved current gain.

As shown in Figure 5c, the integrated LEBJT with the larger emitter area exhibited a linear relationship between the peak EL intensity with respect to the base current intensity under the voltage of Vce = 5 V, demonstrating its electro-optical emitter-like function. Reduced emitter area can provide a higher current gain. An average current gain of 20 and a bandwidth of 180 MHz were obtained by nanorods array.

Overall, the BJT-integrated LED demonstrated some promising advantages such as a compact circuit, reduced power dissipation, and improved energy conversion efficiency by integrating BJTs and LEDs on the same chip. The BJTs acted as current amplifiers to provide precise current control and regulation, significantly reducing base current and providing a current gain of approximately 20 [38], resulting in a good driving and dimming control. Vertically on-chip-integrated BJTs and LEDs can achieve larger device scale and higher resolution and might be one of the solutions for high-brightness and high-resolution display scenarios. However, there are still some problems to be solved. The fabrication of BJT-integrated structures is complex, requiring a special growth process and micro-fabrication technical support, which greatly increases the manufacturing cost and reduces yield [39]. The device size and electrical characteristics of BJTs and LEDs need to be carefully designed and accurately matched to avoid the current mismatch and power loss. The BJT-integrated micro-LED technique is expected to promote the integration and support for the development of display technology and visible light communication.

### 2.2. MOSFET-Integrated Micro-LED

In 2014, a research team from National Taiwan Normal University fabricated MOSFET structures on exposed GaN layers, and connected LEDs in series as depicted in Figure 6a. An LED structure was grown on 2-inch sapphire substrates using a low-pressure metal–organic chemical vapor deposition system. The LED structure was then selectively removed by dry etching (ICP) with Ar/Cl_2_ mixed gases to expose the n-type GaN layer for the subsequent fabrication of MOSFET. Then, an n-type GaN layer on top of the MOSFET mesa was patterned by standard photolithography and ICP dry etching to give a film of 150 nm for the current channel [40]. The monolithically integrated MOSFET devices demonstrated excellent current injection capabilities and were capable of modulating the brightness of the LED by controlling the gate voltage. As can be seen from Figure 6b, the light output power of the LED rapidly increased at low injection currents (less than 20 mA) and reached saturation at higher injection currents (greater than 90 mA). In 2016, Xing Lu and colleagues from the South China University of Technology fabricated an integrated device combining MOSFET with an LED as shown in Figure 7a [41], where they grew p-GaN and n-GaN on top of the LED to realize a p-n-p MOSFET structure. In the process of fabricating the MOSFET, tetramethylammonium hydroxide (TMAH) was employed to repair the sidewall damages of the MOSFET, thereby enhancing its electron mobility and achieving a high output current density exceeding 1.4 kA/cm^2^. The TMAH wet etch was found to be effective in removing damage from the dry-etched GaN sidewall. The I–V characteristic of integrated device indicates that the turn-on voltage is 3 V, and the on-resistance is 4.1 mΩ·cm. Under a forward injection current of 20 mA, the voltage of LED is only 3.72 V. The LED device exhibits a light output power (LOP) of 16 mW and a peak external quantum efficiency (EQE) reaching up to 33%.

In 2016, Kazuaki Tsuchiyama and colleagues from Toyohashi University of Technology demonstrated an integrated device consisting of a silicon-based MOSFET combined with an GaN-based LED as depicted in Figure 8a [33]. They integrated a silicon-based MOSFET structure atop the LED device. They effectively suppressed thermal degradation of the Si-SiO2-GaN structure by optimizing the thickness of the silicon layer and process temperature, all without exceeding 900 °C. This ensured that the integrated component maintained good optoelectronic properties. The I–V curve shown in Figure 8b indicates that the threshold voltage is at 0.8 V, and the micro-LED exhibits an external quantum efficiency (EQE) of 6.5% with a turn-on voltage of 3 V. In 2023, Sang et al. from Nanjing University achieved monolithic integration of GaN-based MOSFET−LED on an LED chip as shown in Figure 9a [42]. The LED wafer was grown on a 2-inch c-plane patterned sapphire substrate by metal–organic chemical vapor deposition (MOCVD). The TJ typically replaces the indium tin oxide (ITO) to serve as a current-spreading layer for LEDs. Meanwhile, the LED and TJ stack present an n/p/n structure, which can also be used for a vertical MOSFET. They realized a stacked n-p-n structure through secondary epitaxial TJ junctions. The micro-LED and MOSFET were interconnected via a conductive n-GaN layer. By modulating the gate voltage, they effectively controlled the injection current into the micro-LED, thus achieving precise control of light output. Figure 9b presents the LOP (light output power) and I–V characteristics of the integrated device, where for a 60-μm micro-LED, the LOP is 0.12 mW (~4.2 W/cm^2^), and the current is 0.3 mA (approximately 10 A/cm^2^). In Figure 9c, the output curve of the MOSFET-LED integrated device exhibits a distinct turn-on threshold that differs from that of a single MOSFET device. Overall, the current density reaches up to 10 A/cm^2^, fulfilling the driving requirements for most micro-LED display applications. This study is based on 2-inch c-plane patterned sapphire and shows the possibility of industrial mass production.

On-chip integration of MOSFETs with LEDs offers significant advantages. The fabrication process for such structures is relatively easy. Without additionally introducing complicated epitaxial structures for transistors, we can directly fabricate the MOSFET on the exposed n-type GaN layer of the LED after dry etching, and serially connect it to the LED through standard semiconductor-manufacturing technologies. Integrating the MOSFET and LED onto the same chip reduces the number and size of external drive circuits, thereby simplifying the layout of the driving circuitry. Integrating the MOSFET in series with the LED also enhances electrical stability and reliability, reducing the risks from electrostatic breakdown and overcurrent damage. Currently, however, due to limitations imposed by the manufacturing processes and material quality, the drive capability of integrated MOSFET devices remains relatively low and is not suitable for high-power micro-LED or large-size display panel requirements.

### 2.3. HEMT-Integrated Micro-LED

GaN-based HEMT has the advantages of high electron mobility, and thus can be used to drive micro-LEDs. A monolithic integrated HEMT-LED device could have a smaller pixel size, accurate current control, higher light output power, and so on. It has high potential in the display and optical communication fields.

In 2016, Liu et al. from Hong Kong University designed and reported horizontally integrated HEMT-LED devices with an optimized GaN/AlN buffer structure, the schematic of which is shown in Figure 10a [34]. The HEMT part on the LED was achieved by selective epi removal (SER) and selective area growth (SAG). A record high forward breakdown voltage of up to 570 V and a reverse breakdown voltage of up to 230 V were achieved by the optimized GaN/AlN buffer layer. The I–V characteristic plot in Figure 10c demonstrates that the HEMT-LED device has an on resistance of 18.5 Ω·mm and a turn-on voltage of about 3 V. It is important to mention that the optimization of growth process will definitely bring great improvement to the performance of integrated devices. Recently, LIU et al. from Southern University of Science and Technology demonstrated a horizontally integrated HEMT-LED device array with a resolution of 20 × 20 [43]. Micro-LED pixels and HEMT cells were acquired by optimized selective area regrowth (SAR). The HEMT-LED device can be illuminated under mA current injection, and the maximum current density can reach 690.4 A/cm^2^. The peak EQE of the integrated device was measured to be 3.36%, and the maximal LOP was 36.2 W/cm^2^, demonstrating a relatively good performance. It can be expected that these high-performance integrated devices could be adopted in areas such as smart watches and VR/AR.

Different methods of integration will also bring different impacts on the device performance. In 2014, Liu et al. from Hong Kong University of Science and Technology fabricated HEMT-on-LED vertically integrated structures as shown in Figure 11a [44]. The AlGaN/GaN HEMT structure was grown on a 2-inch LED epitaxial wafer by MOCVD. The integrated device exhibited a peak transconductance of 244 mS/mm and a maximum output current of 920 mA/mm with an on-resistance of 2.58 Ω-mm. The turn-on voltage of the integrated LED was 3.1 V at an injection current of 10 mA. In 2018, Cai et al. from the University of Sheffield innovatively proposed a horizontally integrated circular HEMT-LED device, as shown in Figure 12a,b [45]. The n-GaN of the LED is connected with the two-dimensional electron gas channel of HEMT to achieve uniform green light emission at the wavelength of 507 nm. The circular structure has better current spreading, where the I–V characteristic shows that the turn-on voltage is ~3 V. The integrated LED can be driven and controlled well by adjusting the gate voltage of the HEMT. In 2019, Chen et al. from South China University of Science and Technology investigated vertically integrated HEMT-LED devices, as shown in Figure 13a,b [46]. An Mg doping process was applied to reduce the interconnect resistance. The output power can reach 80 mW under a forward voltage of 10 V. Its equivalent external quantum efficiency (EEQE) was measured to be 6.15% and the light output power per module area (LOPMA) of the integrated device was 12.5 W/cm^2^. This controllable and uniform emission will play an important role in full-color LED displays.

HEMT-integrated micro-LEDs show good controllability of emission brightness by injecting current, proving their feasibility in a wide range of applications, such as smart watches, HUDs, and AR/VR. GaN-based HEMTs have high-speed switching characteristics and excellent high frequency performance, and thus can provide support for fast current control and regulation, in other words improving the refresh rate of displays. HEMTs have higher current density and power carrying capacity, and can provide higher current drive for LEDs to achieve higher brightness and light output power. Better current spreading can be achieved by optimizing the device structure and material selection. This helps to improve the uniformity and stability of light output. However, the regrowth of HEMTs inevitably brings the problem of thermal and lattice mismatch, increasing the manufacturing costs. We often use enhancement transistors as driving circuits to reduce energy consumption, but it is difficult to fabricate depletion-mode HEMTs.

### 2.4. TFT-Integrated Micro-LED

Typical TFTs cannot meet the increasing needs in micro-LED displays for high density integration and high current density. A new type of TFT is expected to offer a solution in high-resolution displays. Due to the good carrier mobility, crystal quality, and low temperature fabrication, MoS_2_ thin films have been adopted to enhance the electrical properties and current density of TFTs, an can be used as excellent micro-LED drivers. In 2021, Meng et al. from Nanjing University demonstrated an integrated device of MoS_2_ TFT-LED, as shown in Figure 14a [47]. The use of low-temperature ultra-clean growth of a high-quality MoS_2_ single-crystal film gives the TFT good electrical properties. In that work, the drive current of MoS_2_ TFT was 210 μA/μm with good uniformity. Therefore, the blue and green micro-LEDs are extremely bright as driven by the TFT with the gate voltage = supply voltage = bias = 8 V. The integrated device exhibited a response time of ~330 ns, with a single pixel running at 2.4 MHz, which can meet the demands of displays in TV screens, AR/VR, and smartphones. A high-resolution display with 1024 pixels was demonstrated at the system level, of which each pixel consists of one micro-LED and one MoS_2_-TFT.

In 2022, Hwangbo et al. from Yonsei University proposed the direct synthesis of MoS_2_ thin films on GaN-based epitaxial wafers to form TFT arrays [48]. The subsequent monolithic integration of the MoS_2_ TFT with micro-LEDs makes it easy to produce an active-matrix micro-LED display as shown in Figure 15a. This monolithic integration significantly reduced the interconnect delays, increasing the operating frequency. The temperature was kept under 580 °C during the entire fabrication process, avoiding thermal damage to the micro-LED. The proposed method is believed to be applicable to large-scale integration of micro-LED displays.

The integration of TFTs and micro-LEDs is a solution to the problem of the resistance of wafer size. But in consideration of minimizing the interconnect delay that limits the device performance, we need to vertically integrate TFTs with higher carrier mobility and micro-LEDs.

## 3. Conclusions

Overall, the on-chip integration technology of micro-LEDs with driving circuits has made significant breakthroughs. In comparison to traditional methods such as mass transfer and bonding techniques, the novel approach of integrating drivers and display elements on a single chip demonstrates numerous advantages. The drop in threshold voltage from 2.9 to 0.8 V represents a significant reduction in power consumption. The increase in carrier mobility from 10.5 to 1810 cm^2^/(V·S) means a faster response frequency. Furthermore, monolithic integration technology not only plays a vital role in the display domain but also holds immense potential in various other fields, including visible light communication and chip-level optical interconnection [49,50,51]. It is believed that through the efforts of an increasing number of researchers, the performance of integrated devices will be further enhanced, device structures will become more streamlined, and functionalities will grow more diverse. In the near future, on-chip integration technology is proposed to have broader application prospects within the realm of micro-LED displays.

## Figures and Tables

**Figure 1 nanomaterials-14-00511-f001:**
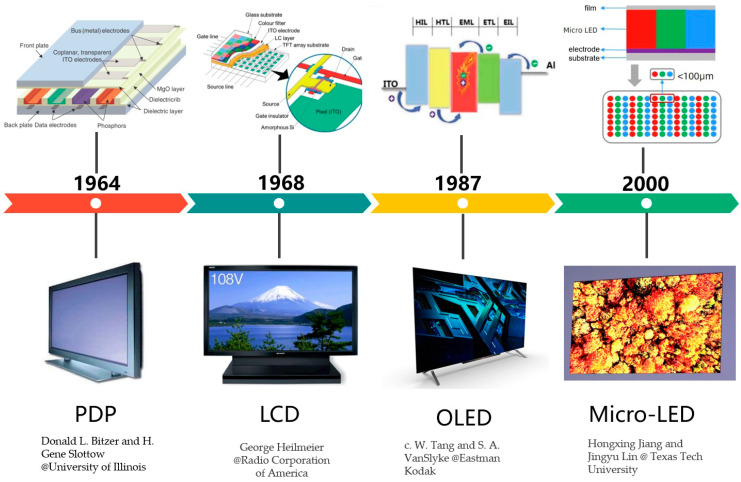
Development of display techniques.

**Figure 2 nanomaterials-14-00511-f002:**
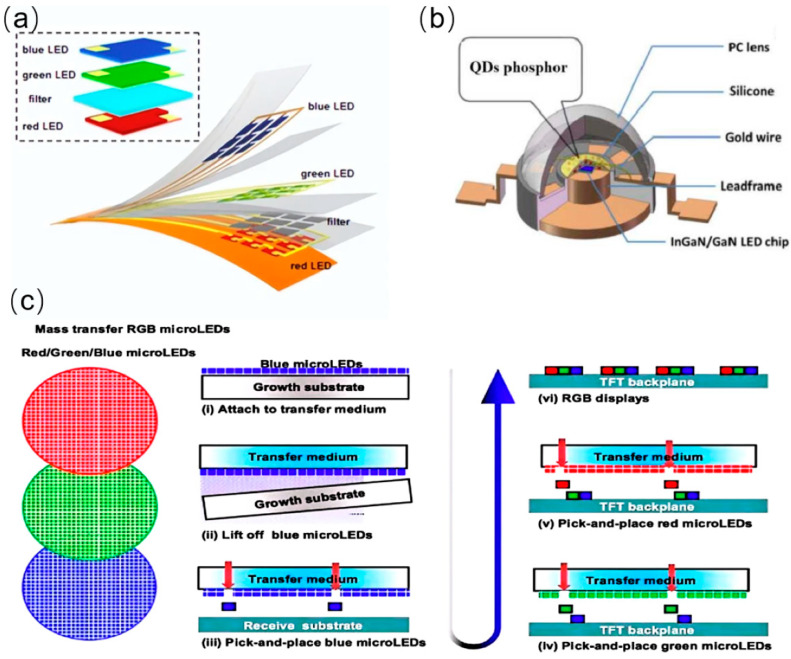
(**a**) Schematic of three-color stacking technology. (**b**) Schematic of a light-emitting quantum dot. (**c**) Schematic of mass transfer.

**Figure 3 nanomaterials-14-00511-f003:**
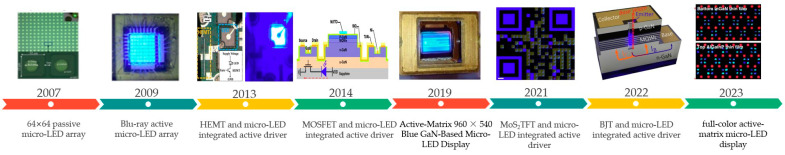
Development of micro-LED driver display.

**Figure 4 nanomaterials-14-00511-f004:**
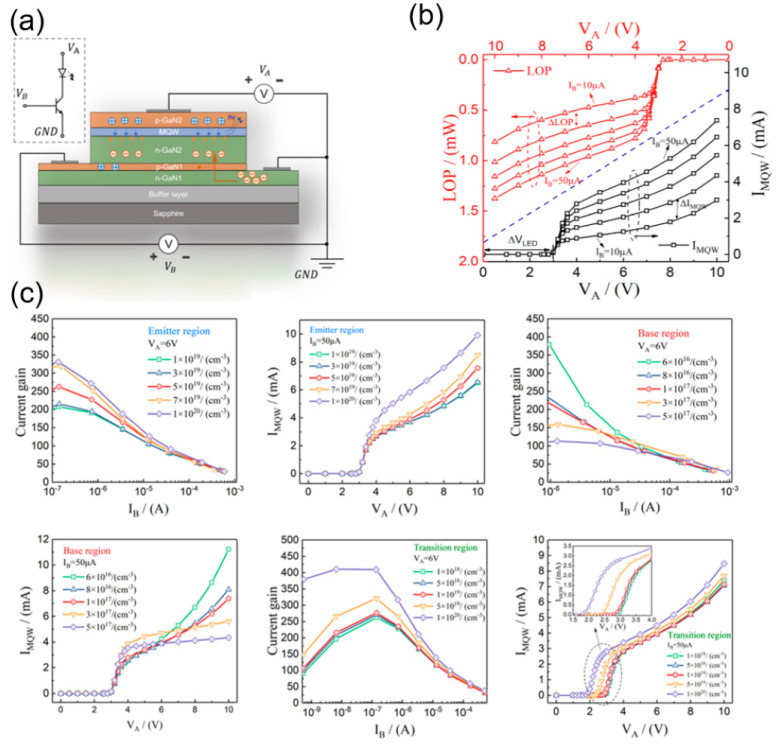
(**a**) Equivalent circuit and operating schematic of the device. (**b**) Output characteristics and optical output power characteristics of the device. (**c**) Current gain and output characteristics for different doping concentrations in the emitter, base, and transition regions [37].

**Figure 5 nanomaterials-14-00511-f005:**
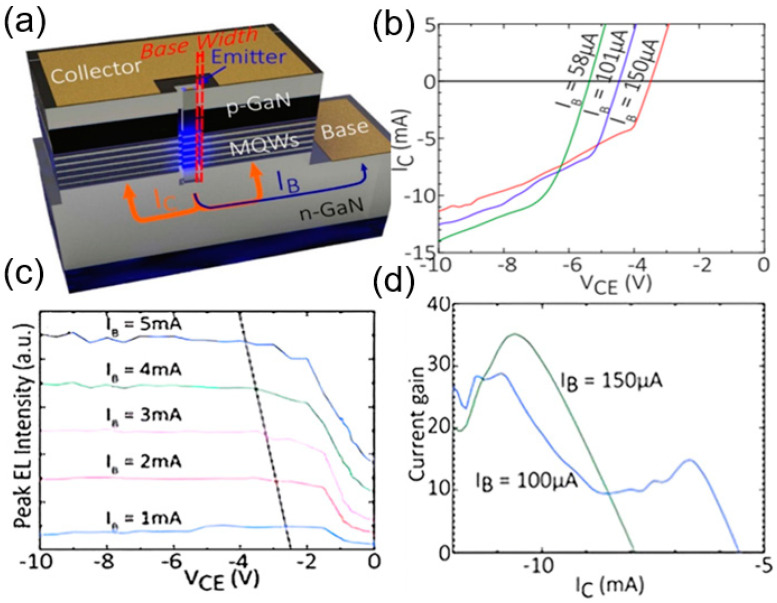
(**a**) 3D schematic of the device cross-section. (**b**) I–V characteristic curves of the emission region of the nanostructures. (**c**) EL mapping of the monolithic BJT structure. (**d**) Current gain extracted from the I–V curves [38].

**Figure 6 nanomaterials-14-00511-f006:**
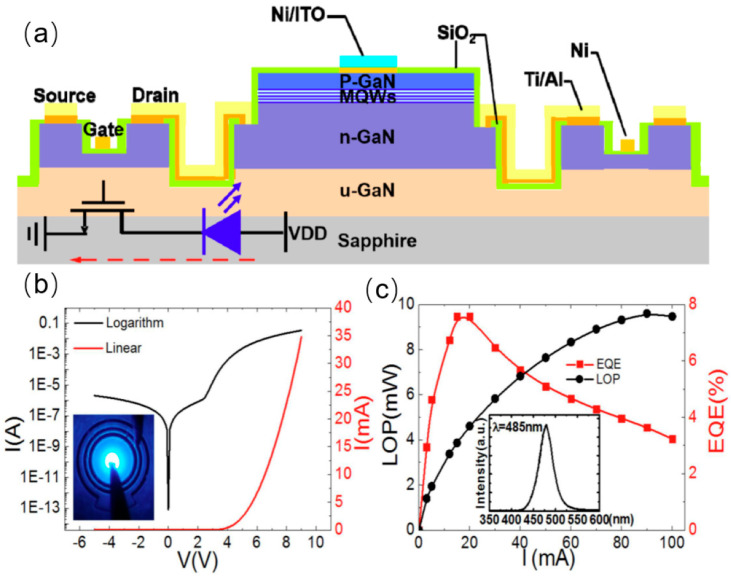
(**a**) Schematic of the MOSFET−LED integrated device and equivalent circuit. (**b**) I–V characteristic curve and picture of illumination under 20 mA current injection. (**c**) LOP and EQE of the integrated device and EL mapping under 20 mA current injection [40].

**Figure 7 nanomaterials-14-00511-f007:**
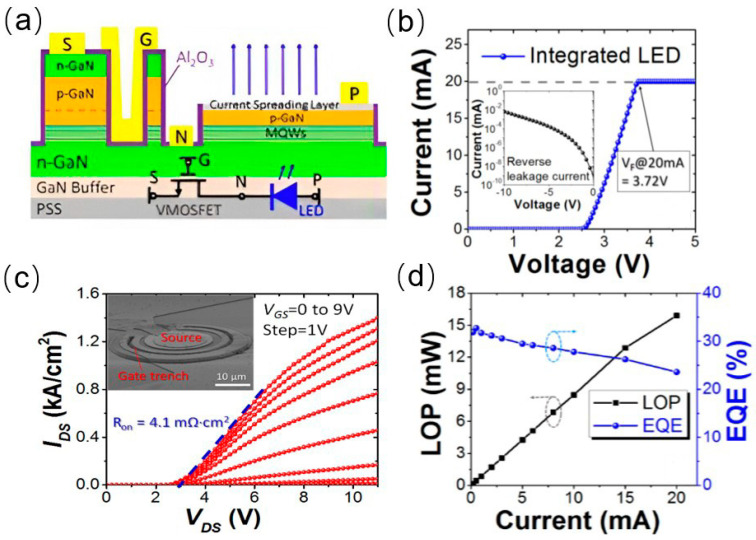
(**a**) Schematic diagram of the integrated VMOSFET−LED. (**b**) I–V characteristic curves of the integrated device. (**c**) Output characteristic curves of the integrated device. (**d**) LOP and I–V characteristics of the integrated VMOSFET−LED device at different drain biases [41].

**Figure 8 nanomaterials-14-00511-f008:**
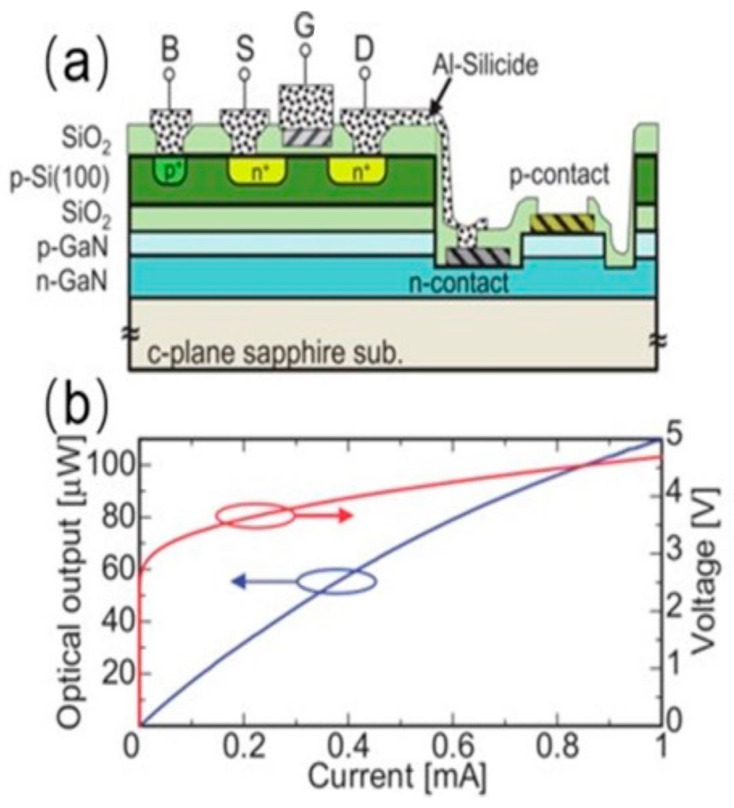
(**a**) Schematic cross-section of the integrated device. (**b**) I–V and output characteristic curves of the integrated device [33].

**Figure 9 nanomaterials-14-00511-f009:**
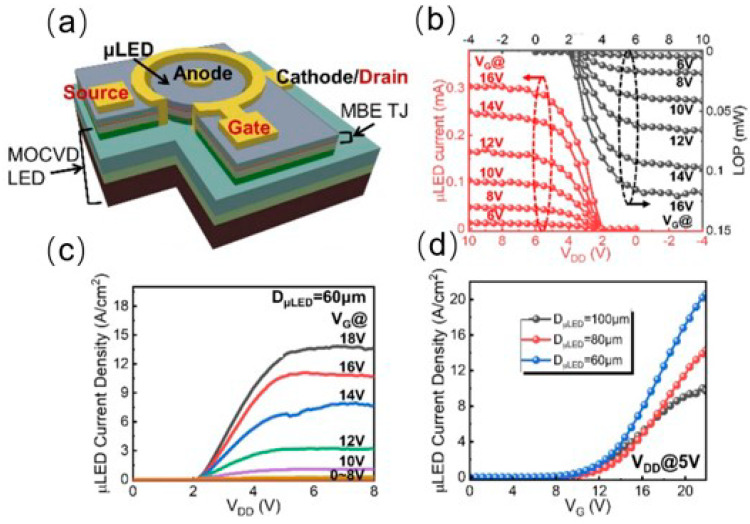
(**a**) Schematic of monolithic integrated LED−MOSFET. (**b**) I–V characteristic curve and LOP of the integrated device. (**c**) Output characteristic curve of the integrated device. (**d**) Transmission curve of the integrated device [42].

**Figure 10 nanomaterials-14-00511-f010:**
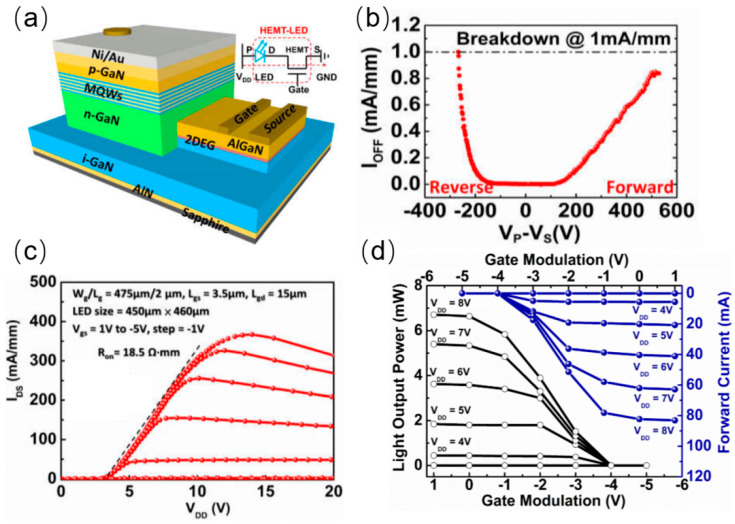
(**a**) Schematic of HEMT−LED device. (**b**) Forward-reverse breakdown voltage. (**c**) I–V characteristics of integrated HEMT−LED. (**d**) Gate bias modulation of optical output power [34].

**Figure 11 nanomaterials-14-00511-f011:**
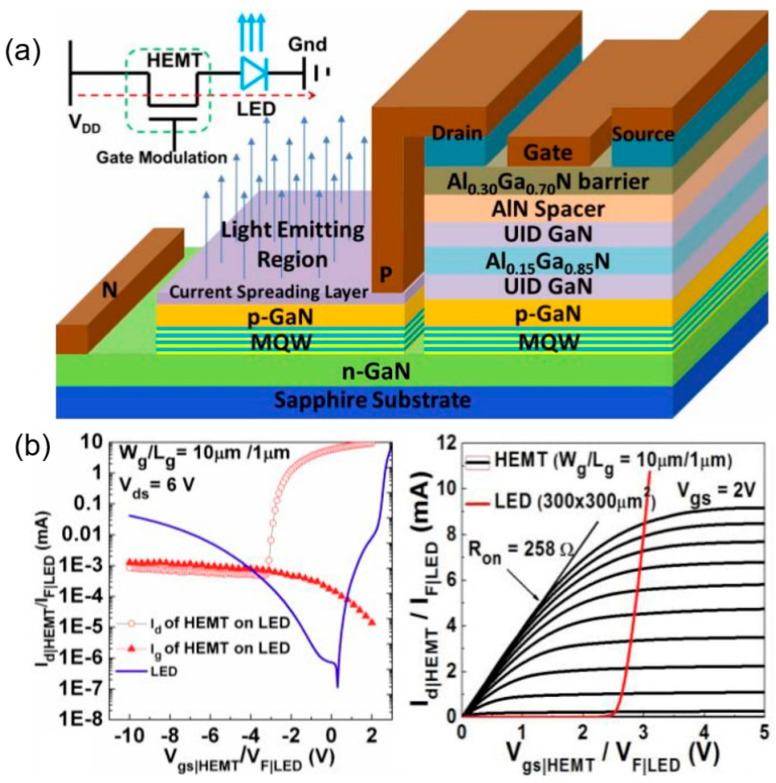
(**a**) Schematic cross-section and equivalent circuit diagram of HEMT−LED. (**b**) Transmission characteristic and output characteristic curves [44].

**Figure 12 nanomaterials-14-00511-f012:**
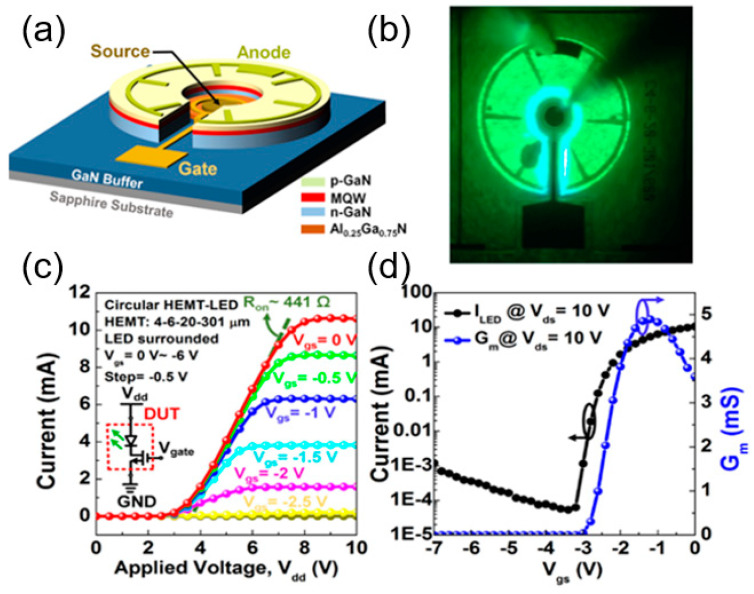
(**a**) Three-dimensional schematic of the device. (**b**) EL lit at 1 mA. (**c**) I–V characteristics of the device and (**d**) transmission characteristics [45].

**Figure 13 nanomaterials-14-00511-f013:**
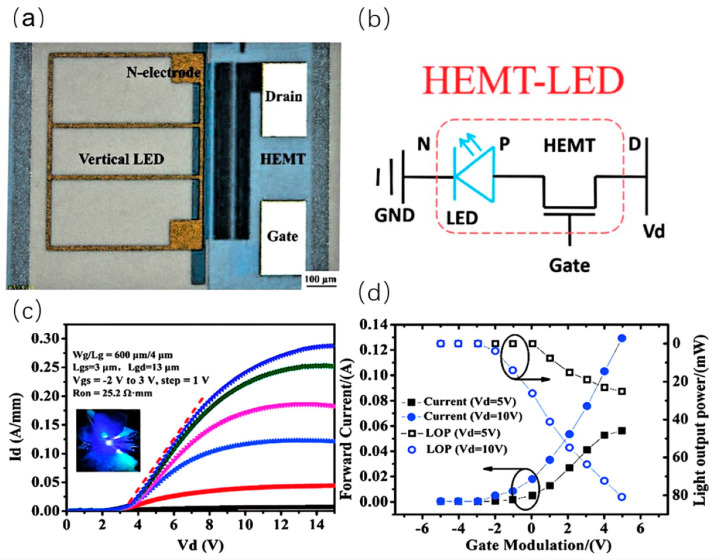
(**a**) Schematic of the integrated HEMT−LED device with vertical structure and (**b**) its equivalent schematic. (**c**) I–V characteristic curves of the integrated device. (**d**) I–V characteristic curves as well as LOP curves of the integrated device modulated by gate bias voltage at 5 V and 10 V drain voltage [46].

**Figure 14 nanomaterials-14-00511-f014:**
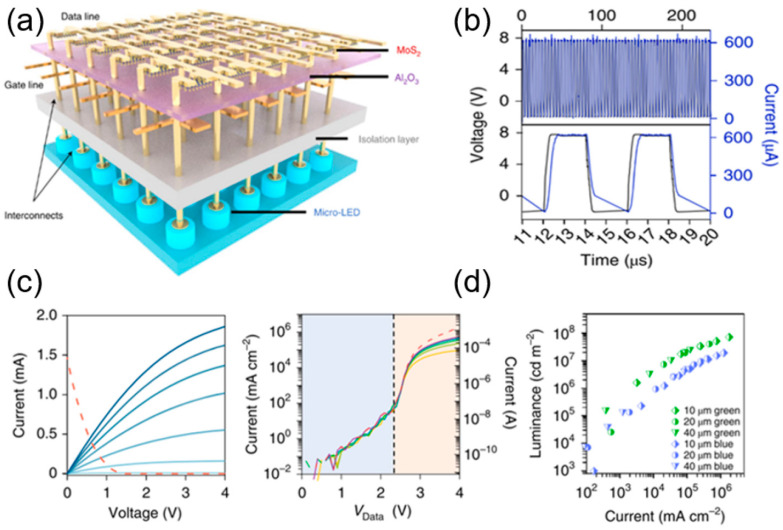
(**a**) Schematic of monolithic integrated TFT−LED integration. (**b**) Stabilized operation at 250 kHz voltage pulse. (**c**) I–V characteristic curves of MoS_2_ tube, micro-LED, and integrated device. (**d**) Brightness of blue-green LEDs [47].

**Figure 15 nanomaterials-14-00511-f015:**
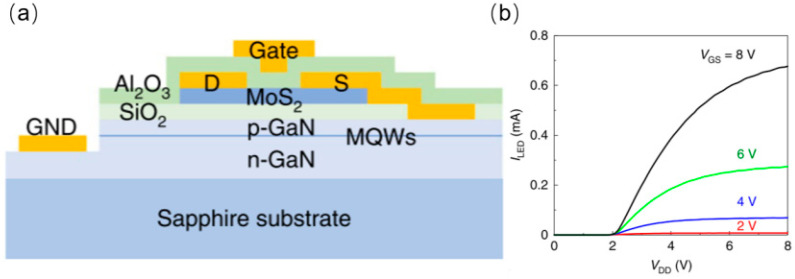
(**a**) Schematic of monolithic integrated TFT−LED. (**b**) I–V characteristics of the integrated device [48].

## Data Availability

All relevant data are available from the corresponding author upon reasonable request.

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
