# Peer review of "Monolithic Integration of GaN-Based Transistors and Micro-LED"

_nanomaterials, 2024, doi:10.3390/nano14060511_

Round 1

Reviewer 1 Report

Comments and Suggestions for Authors

Review

Manuscript Number:  nanomaterials-2897663
Monolithic integration of GaN-based transistors and Micro-LED

In this review paper, the authors have presented recent studies on the new integration methods of  Micro-LEDs with different types of transistors, including the integration with BJT, HEMT, TFT, and MOSFET.

There are some questions and remarks to be answered:

1.    There is a lack of Graphical abstract.

2.    The Abstract should be more specified.

3.    The authors should perform a proof reading of the text (some mistakes, typos, etc.).

4.    The authors should compare the specific technical data, parameters of presented and old methods.

5.    The authors should present also the disadvantages of presented method and show what problems and challenges are to be solved.

6.    In the Conclusions, the authors should indicate what are the specific achievements and advantages of the presented methods through the comparison of the technical parameters of the other methods. The parameters should be given in this chapter and also compared with literature data.

7.    References - the number of literature positions is limited and it should be supplemented, taking into account that this is a review paper.

Comments on the Quality of English Language

The authors should perform a proof reading of the text (some mistakes, typos, etc.).

Reviewer 2 Report

Comments and Suggestions for Authors

While the authors have provided a detailed analysis of various aspects related to the integration of GaN-based transistors and MicroLEDs, there are a few areas that they may have missed addressing:

  1. Cost Considerations: The paper does not delve deeply into the cost implications of implementing GaN-based transistors and MicroLED displays. Cost can be a significant factor in the adoption of new technologies, and a discussion on cost-effectiveness would provide a more holistic view of the integration process.

  2. Environmental Impact: There is limited discussion on the environmental impact regarding the waste products of GaN manufacturing.

  3. Scalability and Production Yield: The authors could have provided more insights into scalability issues and production yield rates associated with the integration of GaN transistors and MicroLEDs. Understanding the challenges in scaling up production and ensuring high yield rates is critical for the widespread adoption of this technology on industrial level.

  4. Long-Term Reliability: While the paper mentions improving efficiency and system stability, a deeper exploration of the long-term reliability of integrated GaN transistors and MicroLEDs would enhance the discussion. Factors such as device lifespan, degradation mechanisms, and reliability under varying conditions are pivotal considerations in real-world applications.

Reviewer 3 Report

Comments and Suggestions for Authors

This review article has been submitted to Nanomaterials. For this reason, I would expect a detailed discussion on how nanotechnology and nanoscopic materials affect the development of GaN-based microLED, but this is missing in the manuscript. Thus, my first comment is:

1) I suggest to include more discussion on nanostructured materials for GaN-based microLEDs

2) Organic materials are quickly mentioned, but they are very important in the GaN-based LED technology. A wider discussion must be included.

3) A discussion of how theory supports the experimental findings could be worthy (modeling of the LED with finite elements, FDTD, ray tracing etc.).

Reviewer 4 Report

Comments and Suggestions for Authors

The paper addresses a comprehensive and very interesting review of the challenge for integrating the driving unit with µLEDs. The main important transistors geometries are presented. My main concern is related to adding technological information, i.e. info about the growth step for reaching integration, which then will clearly evidence the challenges and the difference according to the transistor geometry. Some info is provided on line 296 and line 326 for HEMT integration, and should be added for the others integration transistors.

As regarding the conclusion, authors could emphasis which is the more promising approach between the integration schemes presented in the paper.

I recommend that the paper should be published after additional information has been added, according to the comments listed above and detailed comments listed below.

C1: line 114  define ETC

C2:  line 204 may be could you add some info about the BJT growth , e.g. growth technique, is it on a full wafer? The wafer size, It should be selective growth on localized surfaces, so what about uniformity of the grown layers, are there strains on the limited grown layers? ….

C3: line 217 what does mean “good” ?  

C4: line 229 as for the  BJT growth, can some growth info be added?

C5: Line 238 what technology is used for employing TMAH?

C6: line 254  again how the integration is obtained? Which growth technique  etc….

C7: Line 280  the process is easy: what are the main steps?

C8 line 331 the sentence “The n-GaN of LED connected with the two-dimensional electron gas channel of HEMT to achieve uniform green light emission at the wavelength of 507 nm” is missing a verb.

C9 Line 338 “The output power can reach 80 Mw”  correct Mw  ? could not be Mega !!

Round 2

Reviewer 2 Report

Comments and Suggestions for Authors

Thank you for commenting on the issues with this manuscript! I recommend acceptance!

Reviewer 3 Report

Comments and Suggestions for Authors

The revisions are proper.